# Prediction of Bedform Dimensions on Alluvial Bed in Unidirectional Flow

**Rui Wang * and Guoliang Yu**

SKLOE, KLMIES, MOE, School of Architecture, Ocean & Civil Engineering, Shanghai Jiao Tong University, Shanghai 200240, China; yugl1963@163.com
* Correspondence: w530707596@sjtu.edu.com

**Abstract:** In this study, the bedform dimensions of an alluvial bed in a unidirectional flow were experimentally investigated. A series of flume experiments was conducted; 700 sets of flume and field data were used in developing formulae for predicting the bedform dimensions on alluvial beds in unidirectional flow. Bedform dimensions include the length and height of bedforms generated by the lower and upper flow regimes; the resistance coefficient for the flow in different flow regimes is introduced into the proposed formulae. The momentum boundary-layer thickness was introduced as an independent variable instead of the flow depth. Based on a large amount of flume and field data, the coefficients of each parameter were determined; four typical formulae were used to compare the accuracy of the proposed formulae. The experimental results show that the momentum boundary-layer thickness, hydraulic radius, and resistance coefficient for the flow in different regimes correlate well with the bedform dimensions. The calculation results show that the dimensionless particle size should not be ignored in the calculation of bedform dimensions. The bedform dimensions have an obvious trend of rapid increase with an increase in the ratio of flow depth to sand size ($H/d$). The bedform dimensions obtained using the van Rijn method and the Engelund and Hansen method did not represent the variation trend of the bedform length in the upper flow regime with an increase in $H/d$ when $H/d$ was greater than $10^3$. The calculations using the proposed formulae are more accurate and reasonable than those in previous studies predicting the bedform height and length on an alluvial bed in a unidirectional flow.

**Keywords:** bedform dimensions; flow regimes; momentum boundary-layer thickness; resistance coefficient; alluvial bed

## 1. Introduction

River bedforms are of central importance in many branches of environmental and engineering science for practical management of contemporary river channels [1]. The growth of dunes and larger bar forms leads to significant population displacement and loss of infrastructure and agricultural land [2,3], which increases bed resistance to flows and reduces the sediment transport capacity of the flow. Migration of bedforms may be detrimental to navigation [4], fisheries [5], and submerged structures [6]. Thus, it is important to study the prediction of bedform dimensions in science and engineering.

Bedforms are normally three-dimensional and superimposed structures that develop and decay with flow variance [7], and quantitative description of bedform geometry requires some simplification. The interrelationships and feedback mechanisms between bedform development, sediment transport, and turbulence structures are complex [8]. Even for a steady flow in a straight and uniform flume, bedforms are spatially packed in irregular two-dimensional shapes [9,10]. Some low-amplitude, small-scale bedforms develop over migrating dune bedforms [11,12]. The most important types of bedforms are ripples, dunes, standing waves, and anti-dunes, which are formed by different mechanisms [3]. Their dimensions vary with flow and sediment conditions, and affect the flow and sediment

concentration characteristics [13]. For simplicity, the spatial distribution of bedforms in the lateral direction is typically not considered; bedforms are considered two-dimensional structures (height and length) [14]. Ashley (1990) identifies the principal subaqueous-bedform descriptors as their spacing, their shape (2D or 3D), their hierarchical nature, and the sediment characteristics [15].

The first challenge in regard to bedform dimensions lies in identifying the relevant parameters to describe a bedform train. After the measurements of bedform dimensions in natural channels [16] and the statistical investigation of bedforms [17], a prediction method for bedform dimensions should include the factors of flow and bed sediment. Grain size, water depth, and flow velocity as scaling factors play an important role to controlling the size of subaqueous dunes [7,18]. The formation and disappearance of bedforms occur frequently with the action of flow, which is usually expressed in terms of flow intensity [19]. A laboratory study showed that bedforms appear in the order of sand grains, sand waves, sand dunes, and rushing dunes with an increase in flow velocity [20]. In a uniform and constant unidirectional flow, one or more types of bedforms are concentrated in certain flow conditions after the bed forms, and the flow condition reaches equilibrium [21]. The description for bed sediment is usually expressed in terms of the sediment mixture gradation and specific weight [22]. Dense two-dimensional sand-wave groups are formed in low-lying areas with abundant sediment. In areas with less sediment, discontinuous three-dimensional sand dunes are developed; the shape of the sand dunes is not fully developed [23]. It is generally believed that small ripples are formed in a silty bed, and that a sandy bed is necessary for the formation of large ripples [24]. Researchers have also considered the Froude number [25,26] and hydraulic radius related to bed resistance to predict bedform dimensions [27].

The identified relevant parameters then need to be quantified, providing another level of challenge in terms of appropriate methodologies to be adopted. Based on experimental and field data, dozens of formulae for the height and length of alluvial beds in unidirectional flow have been proposed to predict bedform dimensions, but none were satisfactory. Kennedy and Odgaard (1990) [28] regarded the major formulae as those proposed by Fredsoe (1982) [29], Gill (1971) [30], Raju and Soni (1976) [31], van Rijn (1984) [32], and Yalin (1964) [33]. The formulae proposed by van Rijn (1984) [32] were examined by Julie and Klaassen (1994) [34] and Raslan (1991) [35], and were found to underestimate the bedform height by a factor of two with large transport parameters and overestimate it with smaller transport parameters. The empirical relationship proposed by Flemming (2000) [18] was checked by Santoro et al. (2002) [36] with 1491 couples of bedform height/length but yielded poor matching. Karim (1999) [37] proposed a new method for ripples, dunes, anti-dune/standing waves, and transitional bed regimes; however, he stressed that more research was needed to develop better formulations. In recent years, two novel hybrid intelligence models based on a combination of the group method of data handling (GMDH) and the harmony search (HS) algorithm and shuffled complex evolution (SCE) have been developed to predict bedform dimensions [38]. Artificial intelligence methods are considerably different from the empirical formula of van Rijn (1984) [32]. The difficulty in defining a methodology to adequately quantify bedforms principally arises due to the concomitant deterministic and stochastic natures of bedforms, where sediment beds can further be viewed as series of discrete bedform elements, continuous bed-elevation fields, or some combination of these perspectives [39,40]. It can be seen that the existing method have limitations, and the challenge is to formulate a methodology that realizes sufficient facets to adequately describe sediment waves [41].

In addition, the effect of flow depth on the development of the bedform is still unclear; the ratio of bed sediment particle size to flow depth is typically used as the relative roughness to predict bed dimensions. The parameters in previous research were mostly determined through flume experiments or tests in shallow water, limited to water depths of 0.1~0.5 m, such as the flume experiments conducted by Sami and Hassan (2020) [42] and the test conducted by Qaderi et al. (2018) [38]. The accuracy of existing formulae

for bedform dimensions on alluvial beds in great dimensionless flow depth (the ratio of bed sediment particle size to water depth) is diminished, as sand waves commonly exist in such conditions. For example, there are sand waves with wavelengths of 5~25 m and wave heights of 0.5~2 m in water depths of 132~162 m (particle size of sediment is 0.125~0.188 mm) on the east overseas shelf. The formation of bedforms is only related to the flow motion in boundary layer near the bed; the momentum boundary-layer thickness should be used instead of the water depth to predict the bedform dimensions according to the similarity condition between the experimental and corresponding prototype flows [43].

This study investigates the bedform dimensions generated by unidirectional flow on alluvial beds. Bedforms were divided into lower-regime bedforms (ripples and dunes) and upper-regime bedforms (standing waves and anti-dunes), as bedforms in different flow regimes play different roles in flow resistance [44]. Experiments were conducted firstly, and then the filed data were collected from previous studies. These related parameters, such as momentum boundary-layer thickness, resistance coefficient for flow in different regimes, flow intensity, and hydraulic radius, were used to derive the formulae for bedform length and height. Finally, a comparison with other methods in the conditions of different $H/d$ was conducted. The purpose of this study is to provide a reference for the construction of submarine pipeline burying.

## 2. Methods and Materials

### 2.1. Test Materials

Sands with different median particle sizes were used to configure the testbed in the experiment. Natural sands were dried by dry baking. The drying temperature was 105 °C and the drying time was 24 h. The dried sands were screened through a sieve with an aperture of 10 mm to remove impurities. A TZC-5 particle size analyzer (Shanghai Fangrui Instrument Co., Ltd., Shanghai, China) was used to analyze the sand particles and determine the grain size distribution and median grain size of the sand; an automated sedimentation balance was used. The sand particles measured by the TZC-5 particle size analyzer ranged from 0.038 to 6 mm. The accuracy of the sedimentation balance was 1 mg. The median grain sizes and densities are summarized in Table 1.

**Table 1.** Main parameters of test sediment.

| Sand Samples | Grain Density (g/cm³) | Median Particle size (mm) | Sand Gradation | Sorting Coefficient |
|---|---|---|---|---|
| Natural sand | 2.02 | 0.73 | 1.35 | 2.11 |
| Uniform plastic particles | 1.05 | 3.04 | 1.13 | - |

### 2.2. Experiment Setup

The experiments were carried out in a water flume with a length of 9 m, width of 1 m, and depth of 0.62 m, as shown in Figure 1. Three glass windows were placed on the sidewall of the water flume to read the length and height of the bedforms and measure the velocity of the sand wave. Two 3 kW water pumps were arranged parallel to the left end of the water flume and used to control the flow velocity. The sand thickness in the flume was 12 cm. A rectifier was set at the inlet (right end) of the water flume to stabilize the water flow. The flow velocity in the water flume was measured using a Dop2000 (Doppler ultrasound velocimetry, Signal Processing SA, Lausanne, Switzerland). To expand the experimental range, another water flume with a length of 28 m, width of 6 m, and depth of 1 m was used to observe large-scale bedforms, and the experimental setup was similar. The experimental parameters were measured continuously after the flow in the flume remained uniform and stable.

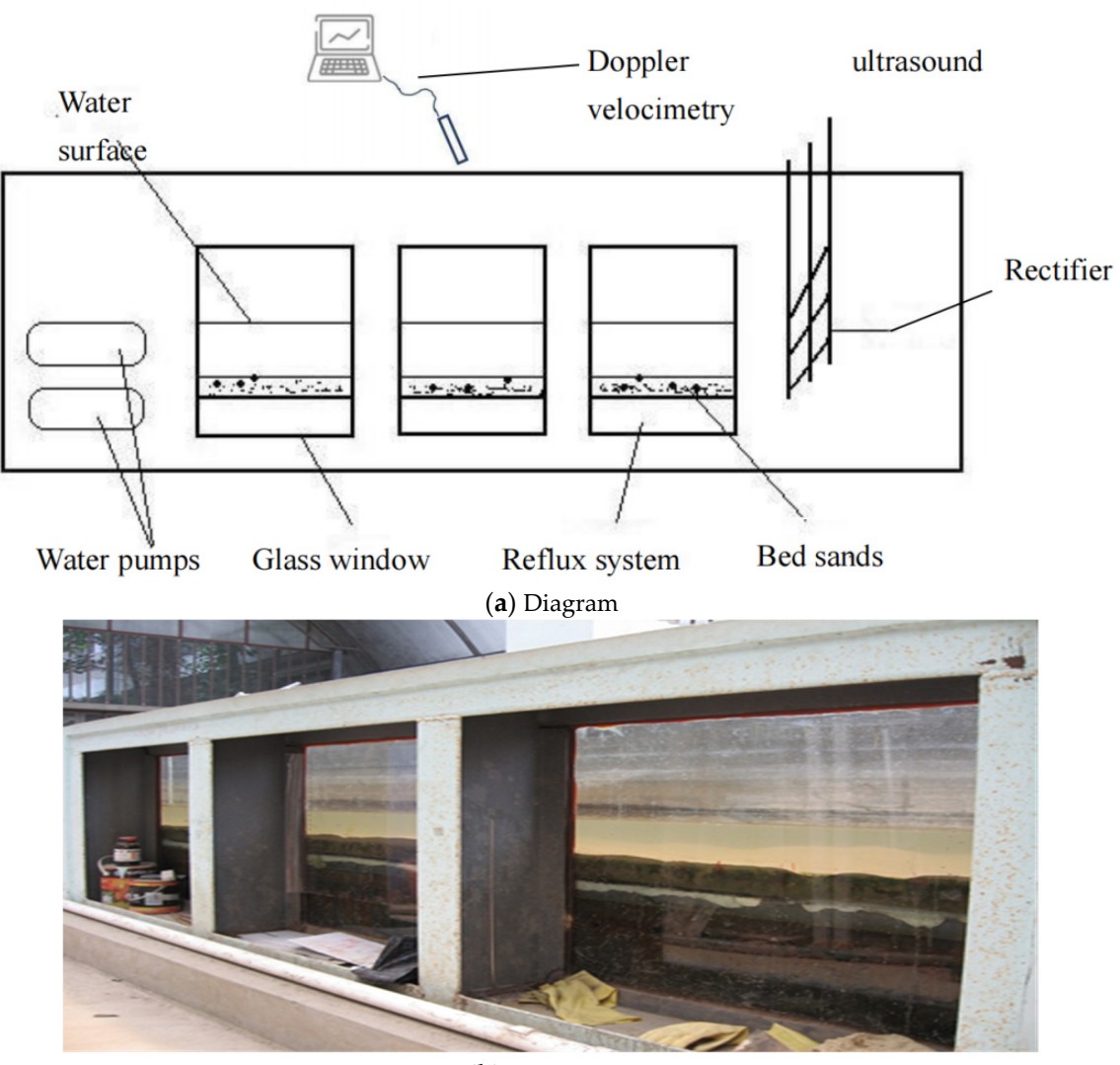

(**a**) Diagram

(**b**) Photograph

**Figure 1.** Experiment setup.

### 2.3. Test Procedure

The bed surface was cleaned, and excess muddy water in the flume was drained. A sand layer with a thickness of 12 cm was laid on the bed surface and leveled. Different amounts of water were injected into the flume to produce different test conditions and water depths according to test requirements. The bed surface was continuously leveled to eliminate unevenness caused by the water-injection process. The water pumps were turned on, and the observation experiments were started. Movement of the sand waves was observed; the parameters were recorded during the experiments.

To reduce the uncertainties produced by manual operation, each set of working conditions was tested 3–5 times; each measurement was performed 3–4 times. The average negative pressure, total wall seepage, and soil plug height were used as the final results. The penetration displacement, time, and velocity were calculated by averaging multiple measurements.

### 2.4. Data Collection

The flow regimes of the movable-bed open-channel flow were divided into three types corresponding to different bedforms, as shown in Table 2. The morphology of a sandy riverbed in uneven conditions is collectively referred to as a sand wave.

**Table 2.** Bedforms in different flow regimes.

| Flow Regimes | Upper Flow Regime | Transition Flow Regime | Lower Flow Regime |
|---|---|---|---|
| Bedform | a. Sand grain<br>b. Sand ridge | Flat bed, from ridge to retrograde ridge | a. Flat bed<br>b. Retrograde ridges and standing waves<br>c. Sharp shoals and deep pools |

When the flow depth is relatively shallow (only tens of centimeters), there are no sand ridges in the development process of the bed surface morphology; however, sand ridges appear with large water depth [18]. Thus, the collected data were grouped into two categories based on the criteria for upper and lower flow regimes proposed by Yu and Lim (2003) [44], in the form of a demarcation line, using the following equation.

$$(1000\ S)q_*^{0.2}\,\sigma_g^{0.2} = \begin{cases} 0.2413\,d_{gr}^2 - 2.385\,d_{gr} + 17.52 & d_{gr} < 14 \\ 3.805\left(d_{gr} - 14\right)^{0.2} + 30.44 & d_{gr} \geq 14 \end{cases} \tag{1}$$

where $q_*$ is the dimensionless flow discharge per unit width; $q_* = q/(gSd_{50}^3)^{1/2}$, in which $q$ is the flow discharge per unit width (l/s); $S = (\gamma_s - \gamma)/\gamma$, in which $\gamma_s$ and $\gamma$ are the specific weights of bed sediment and water, respectively (kg/m$^3$ *g); $\sigma_g$ is the sediment mixture gradation; and $d_{gr}$ is the dimensionless grain size, derived as:

$$d_{gr} = d_{50} \cdot \left(g \cdot S/v^2\right)^{\frac{1}{3}} \tag{2}$$

where $v$ is the kinematic viscosity of water (m$^2$/s); $d_{50}$ is the median particle size of the bed sediment (m); and $g$ is the gravitational acceleration (m/s$^2$).

A total of 698 sets of flume experimental and field-measured data were collected, with 534 sets from a lower flow regime and the remainder from an upper flow regime. A summary of the data is presented in Tables 3 and 4.

**Table 3.** Summary of collected flume experimental data.

| Authors | $Q$ (l/s) | $B$ (m) | $H$ (mm) | $S_0 \times 1000$ | $d_{50}$ (mm) | $\sigma_g$ | $\Delta$ (mm) | $\lambda$ (m) |
|---|---|---|---|---|---|---|---|---|
| Laursen, 1958 [45] | 24.44~181.9 | 0.914 | 76.2~303.3 | 0.55~2.1 | 0.11 | 1.2 | 19.2~33.5 | 0.116~0.155 |
| Stein, 1965 [46] | 84.38~410.58 | 1.29 | 121.9~346.9 | 2.01~3.87 | 0.399 | 1.5 | 48.8~100.6 | 1.372~3.414 |
| Guy et. al, 1966 [47] | 28.32~603.70 | 0.61~2.44 | 57.9~405.4 | 0.15~10.7 | 1.25~2.07 | 1.3~2.07 | 1.5~198.1 | 0.122~6.43 |
| Williams, 1967 [48] | 5.40~37.38 | 0.305 | 28.7~157.6 | 1.06~22.2 | 1.349 | 1.2 | 12.0~50.12 | 0.18~3.260 |
| Vanoni et. al, 1967 [49] | 3.34~185.47 | 0.267~1.1 | 23.2~370.6 | 0.39~2.90 | 0.14~0.23 | 1.38~1.46 | 11.3~50.6 | 0.104~0.23 |
| Williams, 1970 [50] | 1.42~162.34 | 0.076~1.2 | 27.1~222.5 | 0.6~26.2 | 1.349 | 1.2 | 4.0~86.0 | 0.12~3.350 |
| Hung & Shen, 1979 [51] | 351~714 | 2.440 | 296.0~356.0 | 1.21~3.24 | 1.21 | 0.51 | 53.3~94.9 | 0.273~1.74 |
| Wang & Shen, 1980 [52] | 374.1~720.78 | 2.438 | 291.11~353.9 | 0.5755~3.798 | 1.12 | 1.51 | 19.8~54.3 | 1.152~1.77 |
| Termes, 1984 [53] | 101~441 | 1.00 | 168.0~388.0 | 2.737~3.118 | 0.44~0.51 | 1.7 | 62~151 | 1.557~4.76 |
| Brown, 1995 [54] | 125.5~448.75 | 1.756~1.96 | 56.0~356.6 | 1.633~1.847 | 0.8 | 1.3 | 70.6~118.9 | 1.352~1.95 |
| Ayyoubzadeh, 1996 [55] | 4.86~21.14 | 0.400 | 33.6~113.6 | 1.99~2.19 | 0.8 | 1.3 | 3.7~26.2 | 0.68~1.015 |

**Table 4.** Summary of collected field data.

| Rivers | $Q$ (l/s) | $B$ (m) | $H$ (m) | $S_0 \times 1000$ | $d_{50}$ (mm) | $\sigma g$ | $\Delta$ (mm) | $\lambda$ (m) |
|---|---|---|---|---|---|---|---|---|
| Missouri | $8.44 \times 10^5$~$9.83 \times 10^5$ | 154.87~201.11 | 3.26~4.47 | 1295~1740 | 0.2 | 1.35 | 77~282 | 4.6~12.98 |
| Zaire | $1.99 \times 10^7$~$2.85 \times 10^7$ | 913.42~1004.28 | 9.5~17 | 5040~6340 | 0.345 | 1.58 | 1200~1900 | 90~450 |
| Waal | $5.25 \times 10^6$~$6.25 \times 10^6$ | 359.64~429.74 | 8~9.5 | 1357~2019 | 0.48~0.85 | 3.3~3.8 | 700~900 | 9~14 |
| Jamuna | $5 \times 10^6$~$10^7$ | 258.13~813.01 | 8.2~19.5 | 7000 | 0.2 | 1.5 | 800~5100 | 15~251 |
| Parana | $2.5 \times 10^7$ | 641.03~1136.36 | 22~26 | 500 | 0.37 | 1.5 | 3000~7500 | 100~450 |
| Bergsche Maas | $2.16 \times 10^6$ | 146.05~248.28 | 5.8~10.5 | 1250 | 0.18~0.52 | 1.5 | 400~2500 | 6~50 |
| Meuse | $1.16 \times 10^6$~$1.74 \times 10^6$ | 124.6~210.58 | 6.7~9.52 | 1250~1414 | 0.5~0.65 | 0.58~4.81 | 350~850 | 5.5~13.42 |

Notes: $Q$ represents river discharge; $B$ represents channel width; $H$ represents water depth; $\lambda$ represents bedform length; $\Delta$ represents bedform height.

## 3. Results and Analysis

### 3.1. Experimental Results and Data Collection

#### 3.1.1. Experimental Results

The measurements of the flow conditions and bedform dimensions in the water flume with a length of 9 m, width of 1 m, and depth of 0.62 m are presented in Table 5. The test bed was prepared using natural sand with a median particle size of 0.73 mm.

**Table 5.** Flow conditions and bedform dimensions in experiments with a flume width of 1 m.

| No. | Flow Depth (cm) | Mean Velocity (cm/s) | $S_0 \times 10^3$ | Height (cm) | Length (cm) |
|---|---|---|---|---|---|
| 1 | 15.1 | 43.95 | 1.278 | 3.04 | 32.12 |
| 2 | 16.0 | 42.16 | 1.089 | 3.28 | 33.07 |
| 3 | 17.2 | 41.62 | 0.964 | 3.59 | 34.57 |
| 4 | 17.9 | 41.12 | 0.892 | 4.08 | 32.24 |
| 5 | 19.1 | 39.87 | 0.769 | 4.01 | 33.01 |
| 6 | 20.1 | 38.86 | 0.683 | 4.13 | 32.37 |
| 7 | 21.0 | 37.80 | 0.609 | 3.83 | 29.91 |
| 8 | 21.9 | 37.54 | 0.568 | 3.62 | 27.39 |
| 9 | 23.0 | 36.65 | 0.507 | 2.82 | 26.95 |
| 10 | 24.1 | 37.71 | 0.505 | 2.79 | 24.53 |

Note: $S_0$ is the hydraulic slope.

The measurements of the flow conditions and bedform dimensions in a water flume with a length of 28 m, width of 6 m, and depth of 1 m are presented in Table 6. The test bed was prepared using lightweight uniform plastic particles with a median particle size of 3.04 mm.

**Table 6.** Flow conditions and bedform dimensions in experiments with a flume width of 6 m.

| No. | Flow Depth (cm) | Mean Velocity (cm/s) | $S_0 \times 10^5$ | Height (cm) | Length (cm) |
|---|---|---|---|---|---|
| 1 | 28.57 | 8.45 | 3.25 | 2.00 | 69.00 |
| 2 | 28.57 | 9.59 | 4.19 | 2.25 | 82.00 |
| 3 | 28.57 | 10.30 | 4.83 | 2.63 | 90.00 |

#### 3.1.2. Bedform Development

Figure 2 shows the development course of sand waves during the experiments. It is observed that sand waves pass through four periods from an initial state to a stable state. In the initial period, the sand-wave length was short, and the change rate of the sand-wave shape was fast compared with that in other periods, indicating that the sand wave was in an unstable state. In transition period I, the sand-wave length slightly increased compared

with that in the initial period, and there was an obvious superposition of sand waves. The sand-wave shape in transition period II was more stable than in the previous two periods, and an asymmetric waveform was observed. After further development, the sand waves entered the fourth stage. During this period, the sand waveform was stable and the sand-wave length reached its maximum. The sand-wave length and velocity did not change significantly, indicating that the sand-wave movement entered a relatively stable period, as shown in Figure 3.

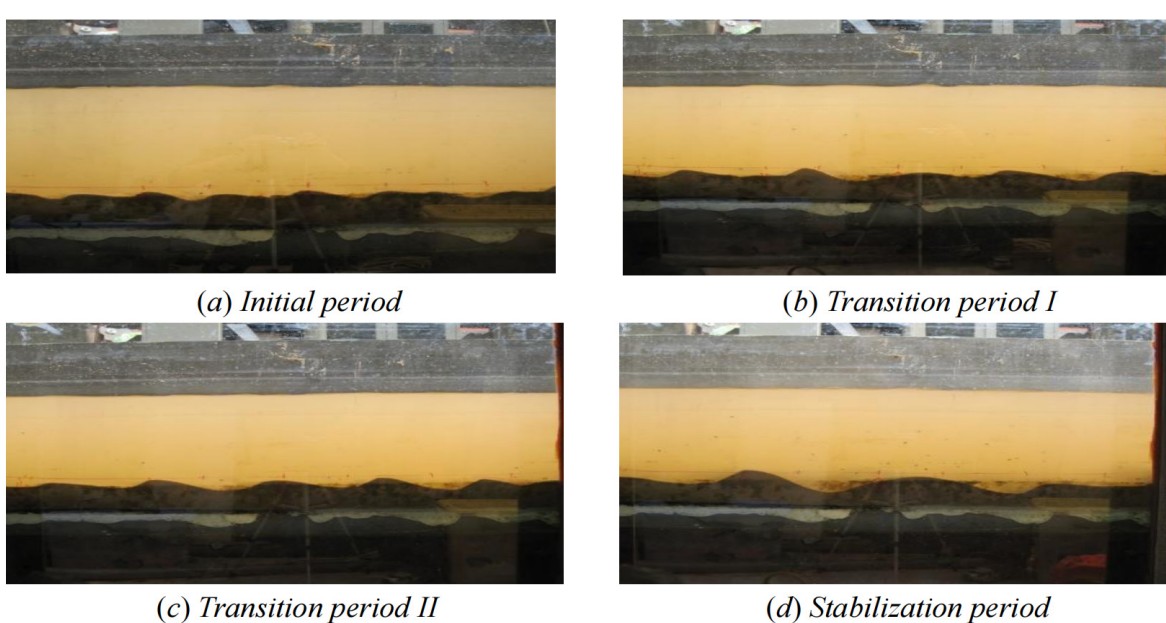

*(a) Initial period*          *(b) Transition period I*

*(c) Transition period II*          *(d) Stabilization period*

**Figure 2.** Stages of sand-wave development.

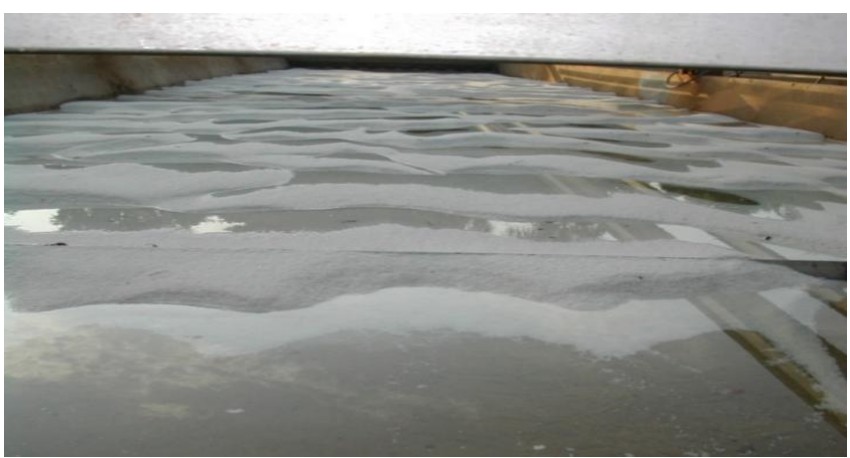

**Figure 3.** Bedforms in the stabilization period.

Sand-wave movement during the experiments is shown in Figure 4a–d, which shows that the sand-wave height increased slowly with time, but the absolute position did not change. The letter b represents parallel movement of the sand waves. The letter c shows the case in which the position of the sand-wave valley does not change, but the position of the wave crest changes. The letter d shows the condition in which the trough of the sand wave is gradually filled and the crest gradually grows over time. These conditions were produced in the experiment; the most common cases were b and d.

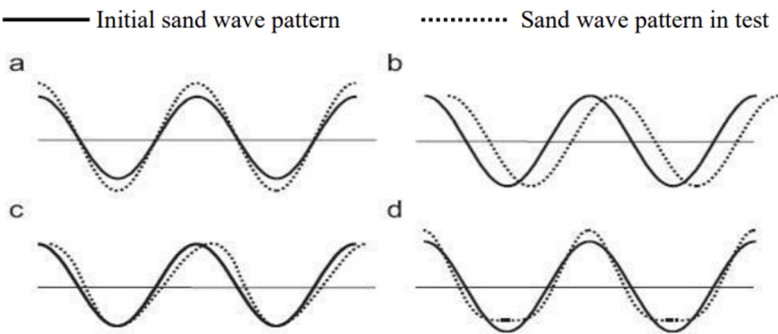

**Figure 4.** Movement of sand waves.

*3.2. Calculation of Bedform Dimensions*

3.2.1. Formula Derivation

The calculation formula for bedform dimensions (bedform height and length) on an alluvial bed in unidirectional flow should be a function of the flow variables and sediment properties; it is assumed that the dimensions and migration of the bedforms are mainly determined by the value of the bed-load transport, which can be described by a dimensionless particle parameter and a transport stage parameter. The calculation formula for the bedform dimensions of an alluvial bed with a high dimensionless flow depth is derived as:

$$\partial(\Delta, \lambda) = f\left(d, \gamma_s, \gamma, \sigma_g, R_b, \theta, \theta_{cr}, \delta, \eta\right) \tag{3}$$

where $\partial$ represents bedform dimensions, including bedform height $\Delta$ and bedform length $\lambda$; $d$ is the bed sediment particle size; $R_b$ is the hydraulic radius associated with bed surface resistance; $\theta$ is the effective flow intensity; $\theta_{cr}$ is the critical Shields number, which represents the incipient condition of the bed sediment; $\delta$ is the resistance coefficient for the flow in different regimes; and $\eta$ is the momentum boundary-layer thickness. Usually, the flow conditions in an alluvial channel are classified as lower flow, transitional flow, and upper flow regimes, as shown in Table 2. The bedform dimensions should contain the length and height of bedforms generated in different flow regimes. The flow resistance mechanism of bed sediment motion is different in different flow regimes; thus, the resistance coefficient for the flow in different regimes is introduced in the proposed formulae. The effect of flow depth on bedform development is still unclear [23]. The sediment motion cannot be calculated directly from the water depth but by the momentum boundary-layer thickness, especially with a large water depth scale [43]; thus, momentum boundary-layer thickness is introduced in the proposed formulae.

Because not all bed shear stress is related to the formation of bedforms, the dimensionless form of the flow intensity was deformed, and the dimensionless variable form of Equation (3) was derived as:

$$\frac{\partial}{d_{50}} = f\left(d_{gr}, \sigma_g, \frac{\theta - \theta_{cr}}{\theta_{cr}}, \frac{R_b}{d_{50}}, \delta, \frac{\eta}{d_{50}}\right) \tag{4}$$

A two-step approximation method was used to improve the calculation precision of the bedform dimensions. The first approximation of Equation (3) can be written as

$$\frac{\partial}{d_{50}} = a_0 d_{gr}^{a_1} \sigma_g^{a_2} \left(\frac{\theta - \theta_{cr}}{\theta_{cr}}\right)^{a_3} \left(\frac{R_b}{d_{50}}\right)^{a_4} \delta^{a_5} \left(\frac{\eta}{d_{50}}\right)^{a_6} \tag{5}$$

where $a_0, a_1, a_2, a_3, a_4, a_5$, and $a_6$ are undetermined coefficients. The relationships between the dimensionless bedform and bed sediment particle size were established with reference to van Rijn (1984) [32]. The calculation methods and analysis for $\theta, \theta_{cr}, R_b, \delta$, and $\eta$ are explained in the next section.

### 3.2.2. Parameter Analysis

(1)     Momentum Boundary-Layer Thickness

Momentum boundary-layer thickness may be a more suitable parameter than flow depth for quantifying the bedform dimensions on the alluvial bed. The bedform dimensions are closely related to the bed sediment motion; the dynamic source is the shear stress generated by the fluid flowing through the bed, which is directly related to the flow velocity gradient. The flow velocity gradient is related to the fluid viscosity and is generated by the bed nonslip condition when the fluid viscosity is not ignored. The flow velocity distribution was uniform when the fluid viscosity was not considered; that is, the fluid was regarded as an ideal fluid. According to the basic theory of boundary layers, the viscosity of the fluid outside the boundary layer should be ignored, and the viscosity of the fluid inside the boundary layer must be considered even if it is low. Thus, the flow velocity gradient was mainly concentrated inside the boundary layer, generating momentum loss in the fluid passing through this region. The fluid momentum loss in the boundary layer is explained by the momentum boundary-layer thickness [43], calculated as:

$$\eta = \frac{\eta^*}{\ln(30.3\frac{\eta^*}{d})} + 2\frac{\eta^*}{\ln^2(30.3\frac{\eta^*}{d})} \tag{6}$$

where $\eta^*$ is the boundary-layer thickness (m), and $d$ is the sediment particle size (m). At the scale of flume experiments or tests in natural shallow water, the boundary-layer thickness is generally determined by the flow depth, $\eta^* = H$.

The relationships between the momentum boundary-layer thickness and bedform dimensions are illustrated in Figure 5, and the field data in lower flow regimes are shown as an example. It is observed that both the bedform height and bedform length in the lower flow regime increase in a power function with an increase in the logarithmic form of the momentum boundary layer.

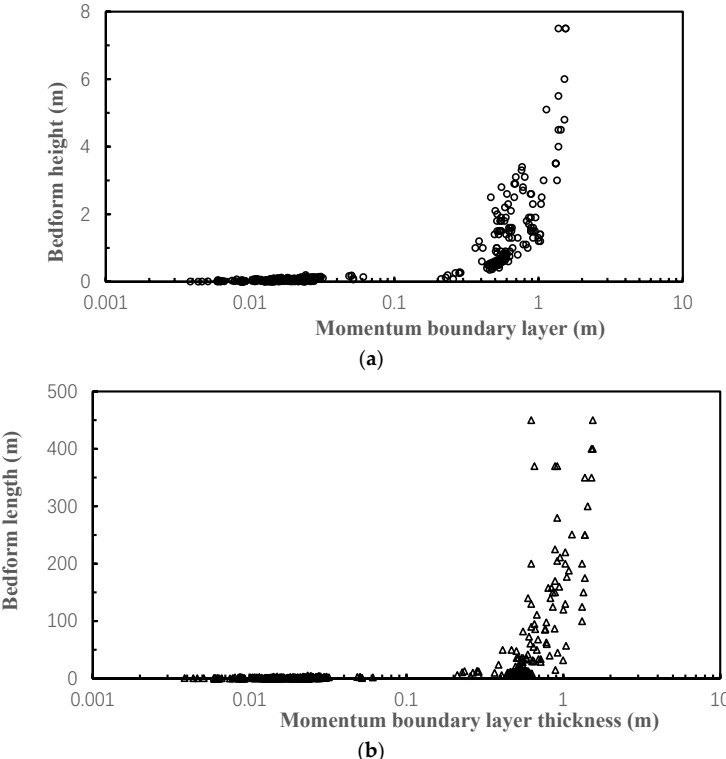

**Figure 5.** Relationship between momentum boundary-layer thickness and bedform dimensions in a lower flow regime. (**a**) Relationship between momentum boundary-layer thickness and bedform height. (**b**) Relationship between momentum boundary-layer thickness and bedform length.

(2)    Resistance Coefficient for Flow in Different Regimes

According to Yu and Lim (2003) [44], the resistance coefficient $\delta$ is correlated with the dimensionless form of bed shear stress $\xi$ ($\xi = ln\theta/\theta_{cr}$), as shown in Figure 6; $\theta/\theta_{cr} \leq 1$ represents a flow intensity lower than the incipient condition of the bed sediment, and the bedform is flat. In the lower flow regime, with an increase in $ln\theta/\theta_{cr}$, the resistance coefficient $\delta_L$ decreased with formation of ripples and dunes. The data points for the resistance coefficient $\delta_L$ and the dimensionless bed shear stress $\xi$ in a lower-energy regime when $1 \leq \theta/\theta_{cr} \leq 250$ are plotted as the red line in Figure 7. These relationships can be expressed as:

$$\delta_L = -0.0044\xi^3 + 0.0661\xi^2 - 0.352\xi + 1 \tag{7}$$

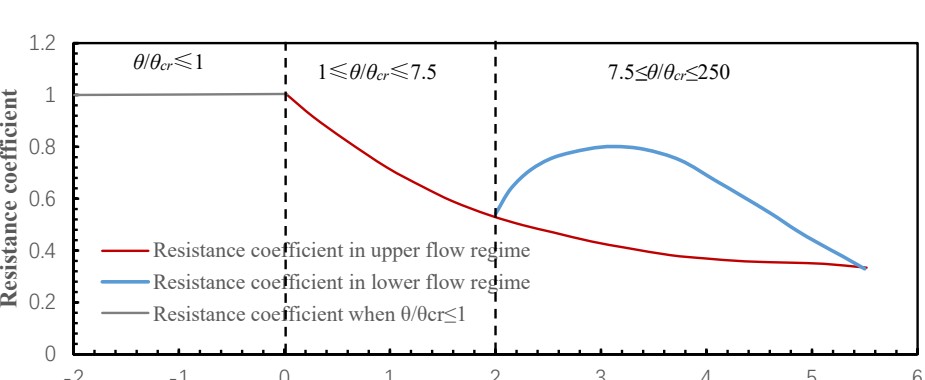

**Figure 6.** Relationship between resistance coefficient and dimensionless form of bed shear stress.

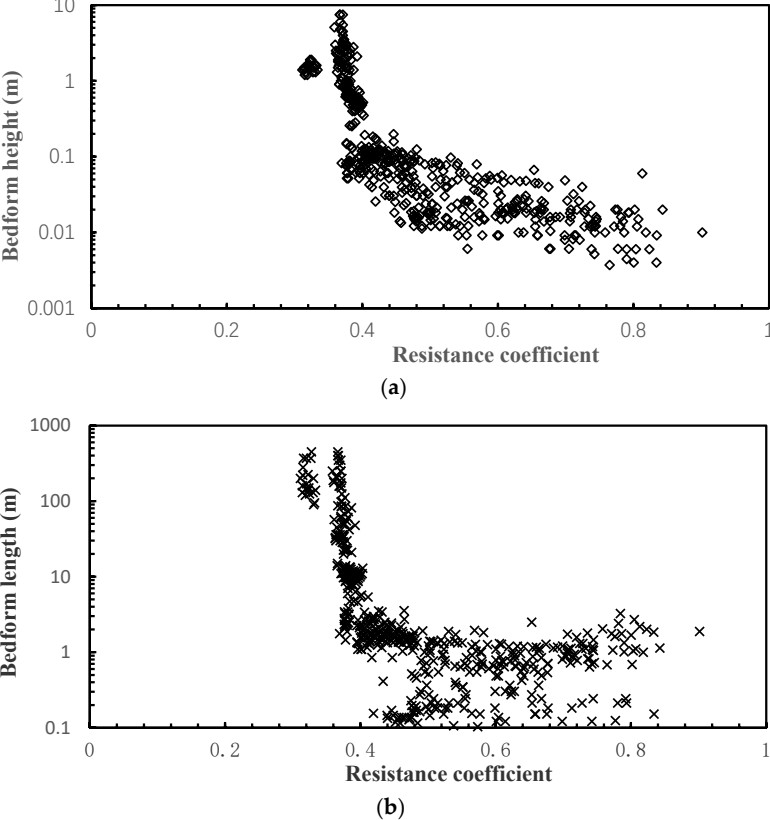

**Figure 7.** Relationship between resistance coefficient and bedform dimensions in lower regimes. (**a**) Relationship between resistance coefficient and bedform height. (**b**) Relationship between resistance coefficient and bedform length.

In the upper flow regime, the relationship between the resistance coefficient $\delta_H$ and $ln\theta/\theta_{cr}$ was divided into two types, as shown in Figure 7. When $1 \leq \theta/\theta_{cr} \leq 7.5$, the relationship can be written as follows, which is consistent with that in the lower flow regime.

$$\delta_H = -0.0044\xi^3 + 0.0661\xi^2 - 0.352\xi + 1 \tag{8}$$

When $7.5 \leq \theta/\theta_{cr} \leq 250$, the relationship can be written as:

$$\delta_H = 0.0337\xi^3 - 0.469\xi^2 + 1.916\xi - 1.644 \tag{9}$$

The relationships between the resistance coefficient for the flow in the lower regime and bedform dimensions are illustrated in Figure 7a,b. It is observed that both the bedform height and bedform length in the lower flow regime decrease in a power function with an increase in the logarithmic form of the momentum boundary layer.

(3)    Flow Intensity

For bedform dimensions on an alluvial bed in unidirectional flow, the effective flow intensity $\theta$ is calculated as:

$$\theta = \frac{\gamma}{\gamma_s - \gamma} \frac{U_*^2}{gd} \tag{10}$$

where $d$ is the sediment particle size, $d = d_{50}$; $U_* = \sqrt{gR_bS_0}$.

The critical Shields number was calculated using the formula proposed by Yu and Lim (2003) [44]. Yu and Lim (2003) [44] rewrote the five equations for $\theta_{cr}$ proposed by van Rijn (1982) [32] into one equation to within $\pm2\%$ error, expressed as:

$$\theta_{cr} = 0.056 - 0.33e^{-0.0115d_{gr}} + 0.1e^{-0.25d_{gr}} + e^{-2d_{gr}} \tag{11}$$

The relationship between $(\theta-\theta_{cr})/\theta_{cr}$ and bedform dimensions in lower flow regimes is illustrated in Figure 8a,b. It is observed that both the bedform height and bedform length in the lower flow regime decrease in a power function with an increase in the logarithmic form $(\theta-\theta_{cr})/\theta_{cr}$.

(4)    Hydraulic Radius

According to the logarithmic law of velocity, the hydraulic radius $R_b$ was calculated using interactions from the implicit equation because the velocity was known.

$$\frac{U}{\sqrt{gR_bS_0}} = 5.75 \log\left(\frac{12.27\chi R_b}{K_s}\right) \tag{12}$$

where $U$ is the mean flow velocity; $K_s$ is the roughness; $K_s = \frac{\sigma_g^2}{d_{50}}$, in which $\sigma_g = \sqrt{\frac{1}{2}\left(\frac{d_{84}}{d_{50}} + \frac{d_{50}}{d_{16}}\right)}$; and $\chi$ is the coefficient with respect to the grain Reynolds number.

$$\chi = 1 - 0.02\frac{1}{\beta^2} + \frac{0.78\beta}{0.02\beta^5 + \beta^2 - 0.8\beta + 1} \tag{13}$$

where $\beta = \frac{U_*K_S}{11.6v}$, in which $v = 1.82e^{-0.027T}$, and T is the temperature in °C.

Similarly, the relationship between the hydraulic radius and bedform dimensions is shown in Figure 9a,b; the field data from lower flow regimes are presented as an example. It is observed that both the bedform height and bedform length in the lower flow regime decrease in a power function with an increase in the logarithmic-form hydraulic radius.

From the analysis, the logarithmic forms of the momentum boundary-layer thickness, resistance coefficient, flow intensity, and hydraulic radius have similar correlations with the bedform height and length in the lower flow regime. These parameters have the same relationship with the bedform height and length in the upper flow regime. Thus, these parameters have an exponential form in Equation (5).

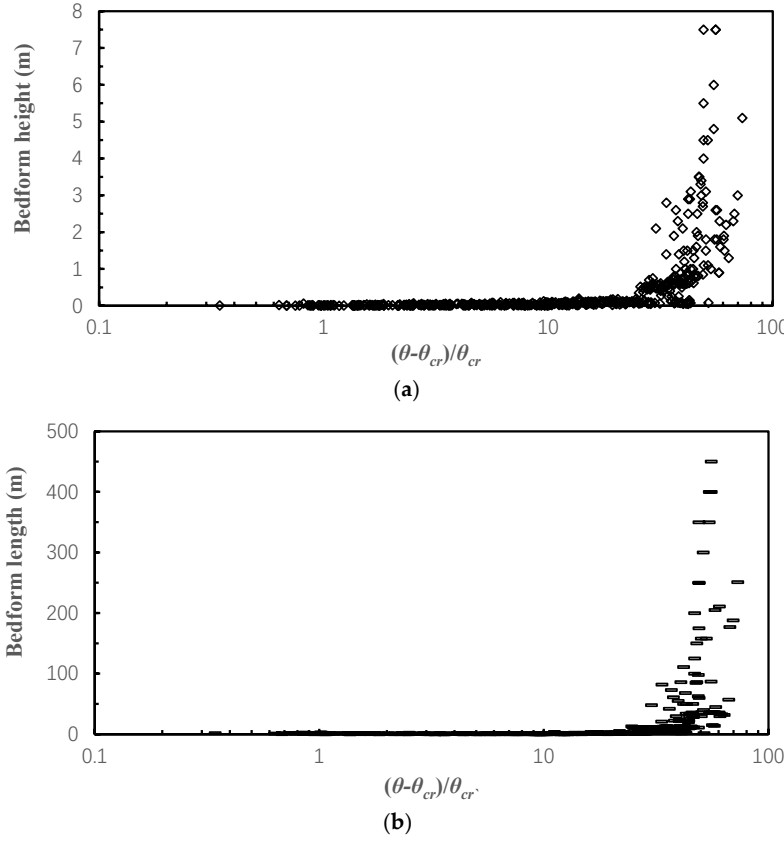

**Figure 8.** Relationship between flow intensity and bedform dimensions in lower regimes. (**a**) Relationship between flow intensity and bedform height. (**b**) Relationship between flow intensity and bedform length.

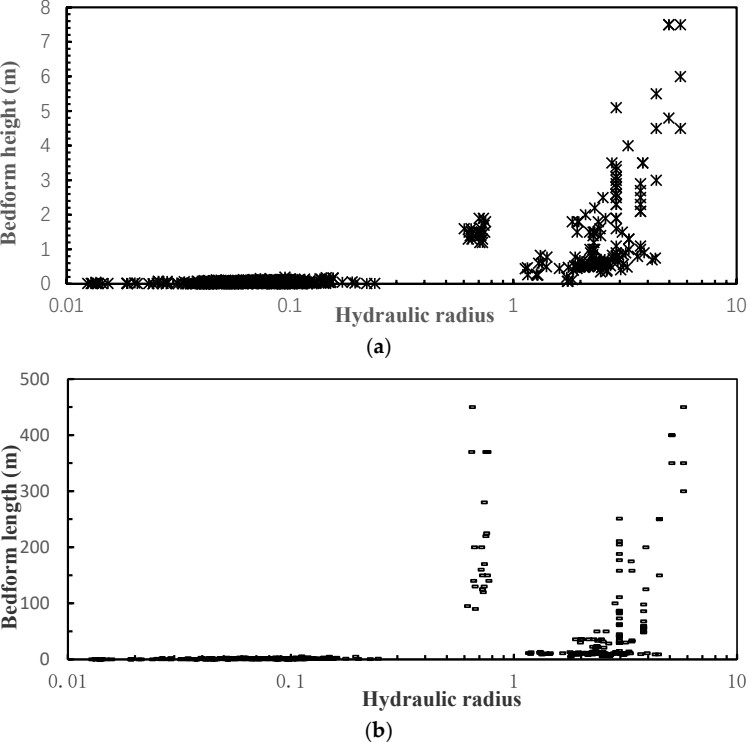

**Figure 9.** Relationship between hydraulic radius and bedform dimensions in lower regimes. (**a**) Relationship between hydraulic radius and bedform height. (**b**) Relationship between hydraulic radius and bedform length.

### 3.2.3. Coefficient Determination

(1)    Bedform Length

The coefficients of Equation (5) for the bedform length in different flow regimes were determined from the test results and collected data using Origin 2021 and Excel 2021. The opposite sides of Equation (5) are logarithms; a multivariate regression method was used to determine these coefficients. The coefficients of Equation (5) are presented in Table 7. The $R^2$ value indicates that the fit of the obtained model is relatively good.

**Table 7.** Coefficients of Equation (4) for bedform lengths in different flow regimes.

| Bedform Type | $a_0$ | $a_1$ | $a_2$ | $a_3$ | $a_4$ | $a_5$ | $a_6$ | $R^2$ | Dataset Number |
|---|---|---|---|---|---|---|---|---|---|
| Lower flow regime | 4.73 | 0.16 | −1.30 | 1.89 | 0.68 | 0.66 | 0.03 | 0.89 | 534 |
| Upper flow regime $(\theta/\theta_{cr} \leq 7.5)$ | 11.12 | −1.32 | −0.40 | 0.05 | $−0.04^2$ | −0.37 | 0.04 | 0.99 | 63 |
| Upper flow regime $(7.5 \leq \theta/\theta_{cr} \leq 250)$ | 4.81 | −1.22 | −0.13 | −0.03 | −0.09 | 0.12 | 0.09 | 0.99 | 118 |

Figure 10 shows the discrepancies in calculated bedform lengths in different flow regimes. In Figure 10a, the discrepancies between the calculated bedform length using the proposed formula and the observed bedform length data in the lower flow regime are presented, and the ratio of the calculation results and observed data is usually around ±50%. The evolution of seabed patterns with changing forcing conditions was out of equilibrium, which may have been the source of the calculation error. With changes in flow conditions, rippled beds do not instantaneously respond to forcing conditions in field and laboratory experiments [56,57]. Figure 10b,c show the discrepancies between the bedform heights calculated using the proposed formula and the observed bedform lengths in the upper flow regime when $\theta/\theta_{cr} \leq 7.5$ and $7.5 \leq \theta/\theta_{cr} \leq 250$, respectively. Almost all of the data are within the 50 percent error line.

(2)    Bedform Height

The coefficients of Equation (5) for the bedform heights in different flow regimes were also determined from the test results and collected data using Origin 2021 and Excel 2021. The same formal deformation of Equation (5) was performed. The coefficients are presented in Table 8; the $R^2$ value indicates that the fit of the obtained model is relatively good.

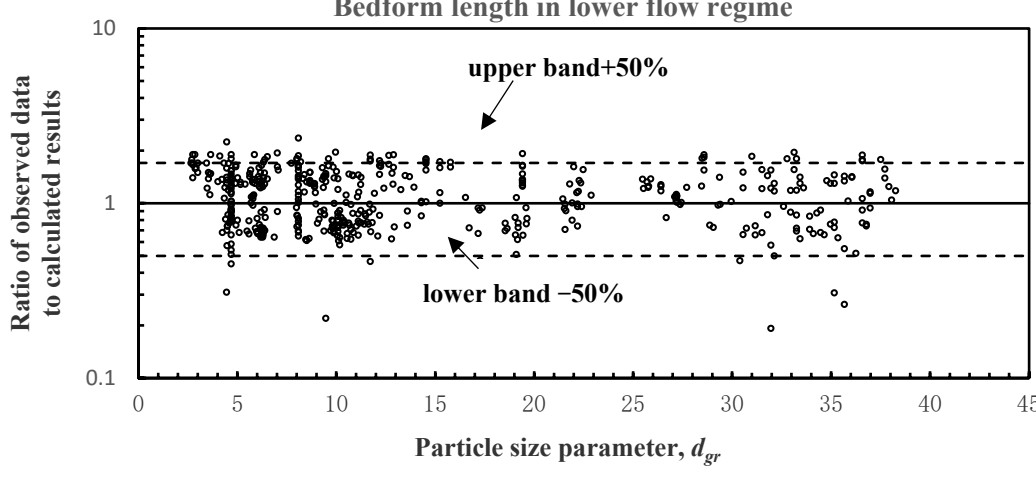

(**a**) Discrepancies in calculated bedform lengths in lower flow regimes.

**Figure 10.** *Cont*.

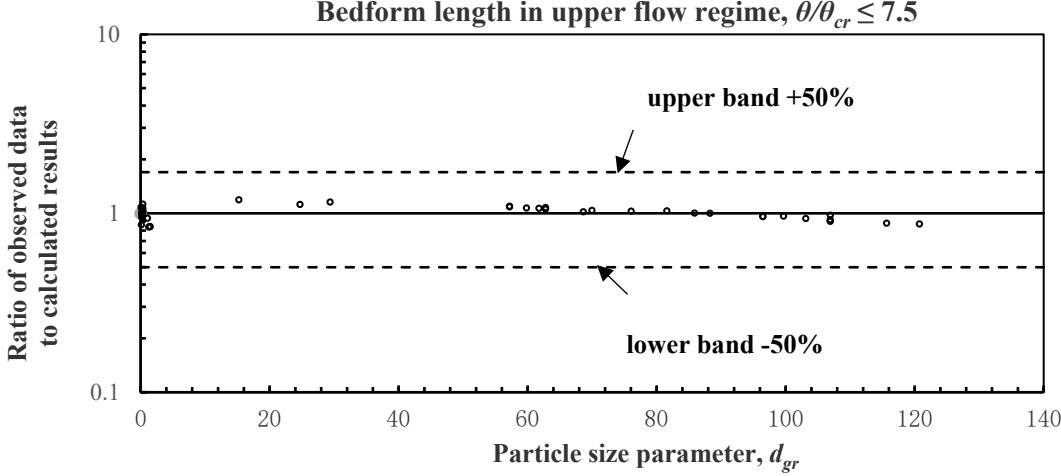

(**b**) Discrepancies in calculated bedform lengths in upper flow regimes when $\theta/\theta_{cr} \leq 7.5$.

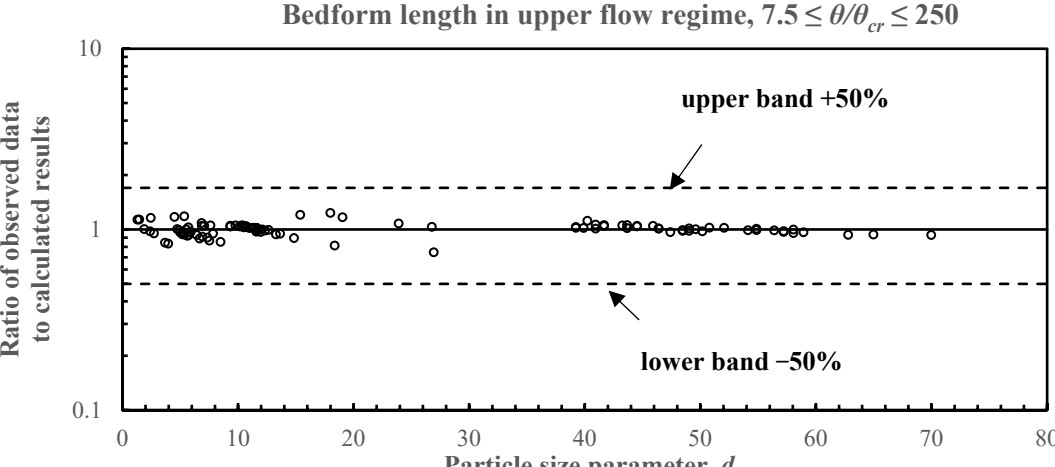

(**c**) Discrepancies in calculated bedform lengths in upper flow regimes when $7.5 \leq \theta/\theta_{cr} \leq 250$.

**Figure 10.** Discrepancies in calculated bedform lengths in different flow regimes.

**Table 8.** Coefficients of Equation (5) for bedform heights in different flow regimes.

| Bedform Type | $a_0$ | $a_1$ | $a_2$ | $a_3$ | $a_4$ | $a_5$ | $a_6$ | $R^2$ | Dataset Number |
|---|---|---|---|---|---|---|---|---|---|
| Lower flow regime | −0.02 | −0.11 | −0.52 | 0.01 | 0.18 | −2.92 | 0.54 | 0.94 | 534 |
| Upper flow regime $\theta/\theta_{cr} \leq 7.5$ | 1.23 | 0.09 | 2.92 | −0.43 | −0.04 | 0.06 | 0.95 | 0.96 | 63 |
| Upper flow regime ($7.5 \leq \theta/\theta_{cr} \leq 250$) | −0.08 | −0.85 | −1.80 | 0.26 | −0.11 | 0.48 | −0.10 | 0.89 | 118 |

Figure 11 shows the discrepancies in calculated bedform heights in different flow regimes. In Figure 11a, the discrepancies between the bedform heights calculated using the proposed formula and the observed bedform height in the lower flow regime are exhibited, and the ratio of the calculation results and the observed data is usually around ±50%. Figure 11b,c show the discrepancies between the bedform heights calculated using the proposed formula and observed bedform heights in the upper flow regime when $\theta/\theta_{cr} \leq 7.5$ and $7.5 \leq \theta/\theta_{cr} \leq 250$, respectively. Almost all data are within the 50 percent error line. It can be seen that the proposed method has good precision.

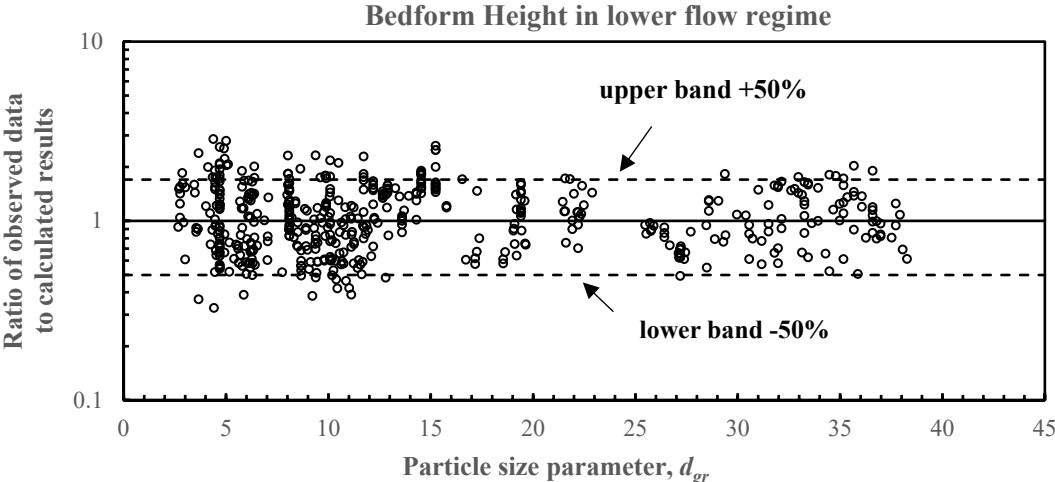

(**a**) Discrepancies in calculated bedform heights in lower flow regimes.

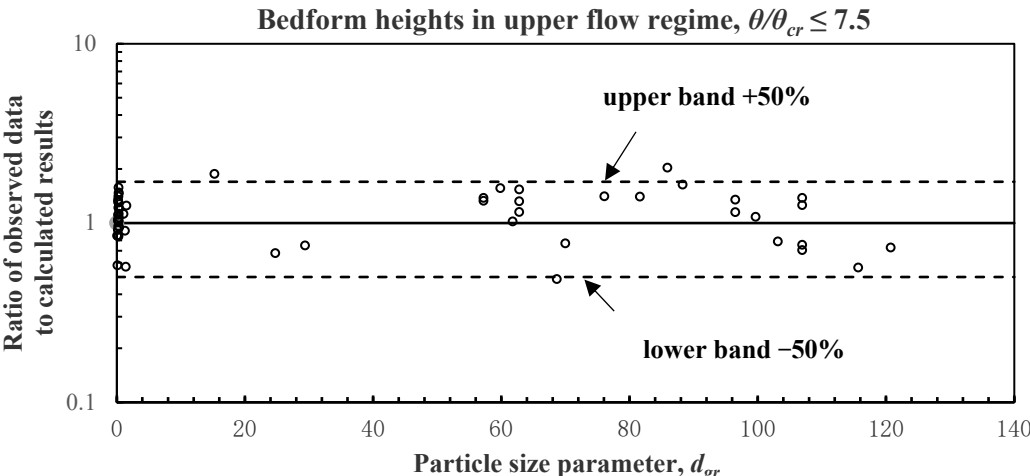

(**b**) Discrepancies in calculated bedform heights in upper flow regimes when $\theta/\theta_{cr} \le 7.5$.

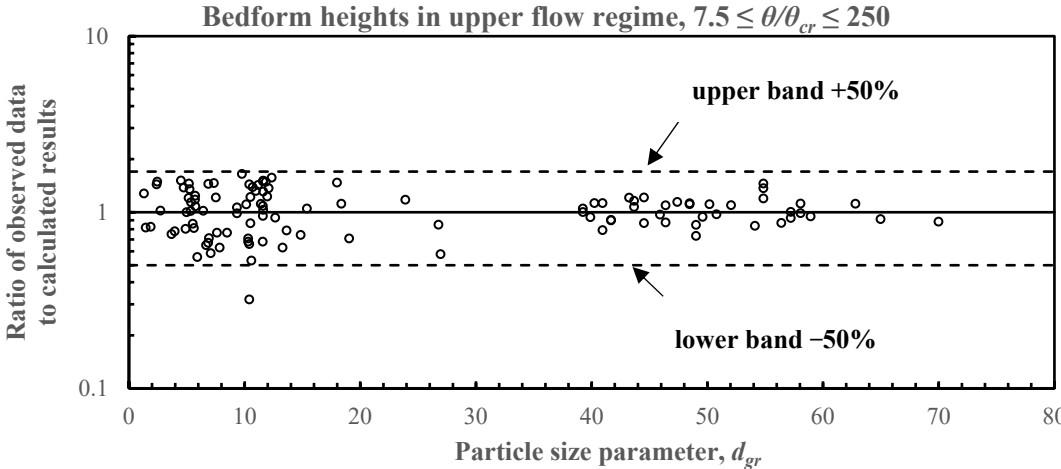

(**c**) Discrepancies in calculated bedform heights in upper flow regimes when $7.5 \le \theta/\theta_{cr} \le 250$.

**Figure 11.** Discrepancies in calculated bedform heights in different flow regimes.

(3)  Influence of Dimensionless Sediment Particle Size

The correlation between the dimensionless sediment particle size and bedform dimensions was weak, as shown in Table 9 and consistent with van Rijn (1984) [33]. The correlation between the dimensionless sediment particle size and bedform dimensions was evaluated using the Pearson correlation coefficient. In statistics, the Pearson correlation coefficient reflects the linear correlation between two variables. All analyses were conducted using SPASS and Excel 2020 software.

**Table 9.** Correlation between $d_{gr}$ and bedform dimensions.

| Flow Regime<br>Flow Intensity | Lower Flow Regime<br>- | | Upper Flow Regime | | | |
| --- | --- | --- | --- | --- | --- | --- |
| | | | $\theta/\theta_{cr} \leq 7.5$ | | $7.5 \leq \theta/\theta_{cr} \leq 250$ | |
| Parameter | $\Delta$ | $\lambda$ | $\Delta$ | $\lambda$ | $\Delta$ | $\lambda$ |
| Pearson coefficient | $-0.29$ | $-0.23$ | $-0.71$ | $-0.17$ | $0.29$ | $-0.54$ |

Nevertheless, the dimensionless sediment particle size is still considered to play an important role in calculating the bedform dimensions and should be included in the proposed formulae. In Table 10, the values of $a_1 d_{gr}$ are always within 2 for the calculation of the bedform length and height in lower flow regimes and the bedform height in upper flow regimes. The maximum value of $a_1 d_{gr}$ reaches 18.4 in the calculation of the bedform length when $\theta/\theta_{cr} \leq 7.5$. This approach is also reasonable, as there was a clear influence of the dimensionless sediment particle size observed in the Missouri River data used by van Rijn (1984) [33] that was not considered; the formulae proposed by van Rijn (1984) [33] were proven to underestimate or overestimate the bedform height at different transport stages.

**Table 10.** Variation in $a_1 d_{gr}$ in different flow regimes.

| Flow Regime<br>Flow Intensity | Lower Flow Regime<br>- | | Upper Flow Regime | | | |
| --- | --- | --- | --- | --- | --- | --- |
| | | | $\theta/\theta_{cr} \leq 7.5$ | | $7.5 \leq \theta/\theta_{cr} \leq 250$ | |
| Parameter | $\lambda$ | $\Delta$ | $\lambda$ | $\Delta$ | $\lambda$ | $\Delta$ |
| $a_1 d_{gr}$ | 0.67~0.89 | 1.17~1.79 | 0.02~18.4 | 0.82~1.53 | 0.06~1.64 | 0.03~1.86 |

### 3.3. Comparison with Other Methods

The calculation formula proposed in this study was compared with commonly used calculation models based on the measured and collected data. The following four commonly used calculation models were used.

(1)  van Rijn method

van Rijn (1984) [33] conducted regression analysis on 84 groups of experimental data with sediment particle sizes ranging from 190 to 23,000 um and 22 groups of field data with particle sizes ranging from 490 to 3600 um; the water depths were greater than 0.1 m. He proposed a relationship between bedform dimensions based on experimental data, derived as:

$$\frac{\Delta}{H} = 0.11 \left( \frac{d_{50}}{H} \right)^{0.3} \left( 1 - e^{-0.5T} \right) (25 - T) \tag{14}$$

$$\frac{\Delta}{\lambda} = 0.015 \left( \frac{d_{50}}{H} \right)^{0.3} \left( 1 - e^{-0.5T} \right) (25 - T) \tag{15}$$

where $T$ is a transport stage parameter, derived as:

$$T = \left( \frac{U'_*}{U_{*cr}} \right)^2 - 1 \tag{16}$$

where $U'_*$ is the bed-shear velocity related to the grains; $U'_* = (g^{0.5}/C')*U$, in which $C' = 18log(12R_b/3D_{90})$ = Chezy coefficient-related grains; $U$ is the mean flow velocity; and $U_{*cr}$ is the critical bed shear velocity according to Shields.

(2)     Engelund and Hansen Method

Garde and Albertson proposed that bedform dimensions are mainly related to the bed-shear stress and Reynolds number of sediment particles at the sand grain stage, and to the bed shear stress and Froude number at the sand ridge stage. Although the experimental data used by Garde and Albertson were insufficient to fully determine this relationship, this attempt was a step forward. Based on limited available information, Hansen obtained the following relationship.

$$\frac{\lambda}{H}S_0\theta^2 = 0.037\left(\frac{U}{\sqrt{gH}}\right)^{5.4} \tag{17}$$

(3)     Raju and Soni Method

According to the relationship between sand-wave movement and the transport rate of bed sediment, Raju and Soni (1976) [31] established a formula for bedform height based on previous laboratory data and field data from the Luznice River. The mean sediment particle size of the previous laboratory data ranged from 0.1 to 1.35 mm; the mean sediment particle size of the field data was 2.40 mm.

$$\frac{\Delta}{d_{50}}\left(\frac{\gamma}{\gamma_s - \gamma}\right)^{1/2}\left(\frac{R_b}{d_{50}}\right)^{1/2}\left(\frac{U}{\sqrt{gR_b}}\right)^4 = 6.5 \times 10^3\theta^{8/3} \tag{18}$$

(4)    Wuhan Hydropower Institute Method

The Sediment Transport Laboratory at the Wuhan Hydropower Institute proposed a simple relation based on laboratory and field data in China.

$$\frac{\Delta}{H} = 0.086\left(\frac{U}{\sqrt{gH}}\right)\left(\frac{H}{d_{50}}\right)^{1/4} \tag{19}$$

The proposed method and these four methods were tested against the data in Tables 3 and 4. The discrepancies between the calculated bedform dimensions and field data are depicted in Figures 12 and 13. The proposed method yielded better predictions than the other methods. The Engelund and Hansen method yielded the poorest prediction for bedform length. The Wuhan Hydropower Institute method produced a better prediction for bedform height than the van Rijn method and Raju and Soni method.

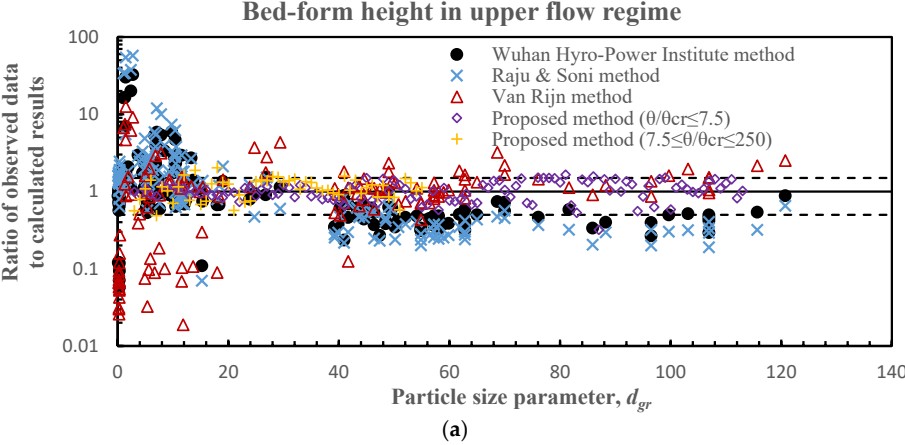

**Figure 12.** *Cont.*

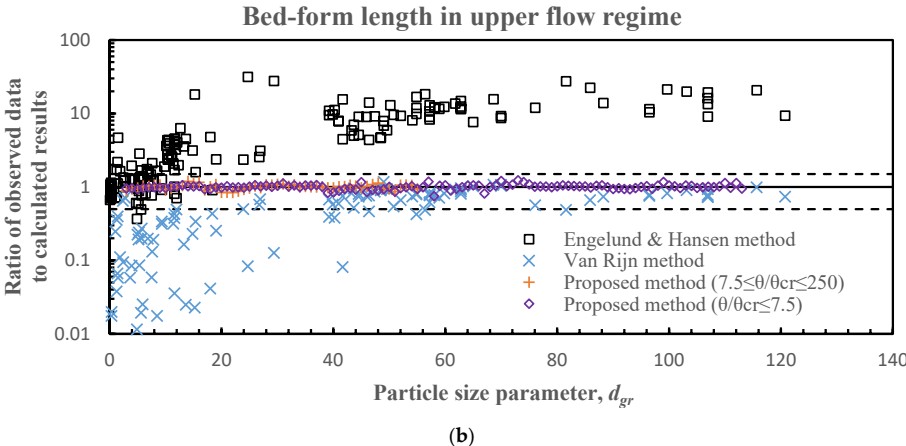

**Figure 12.** Comparison of bedform dimensions in upper flow regimes using different methods. (**a**) Comparison of bedform height observed using different methods. (**b**) Comparison of bedform length observed using different methods.

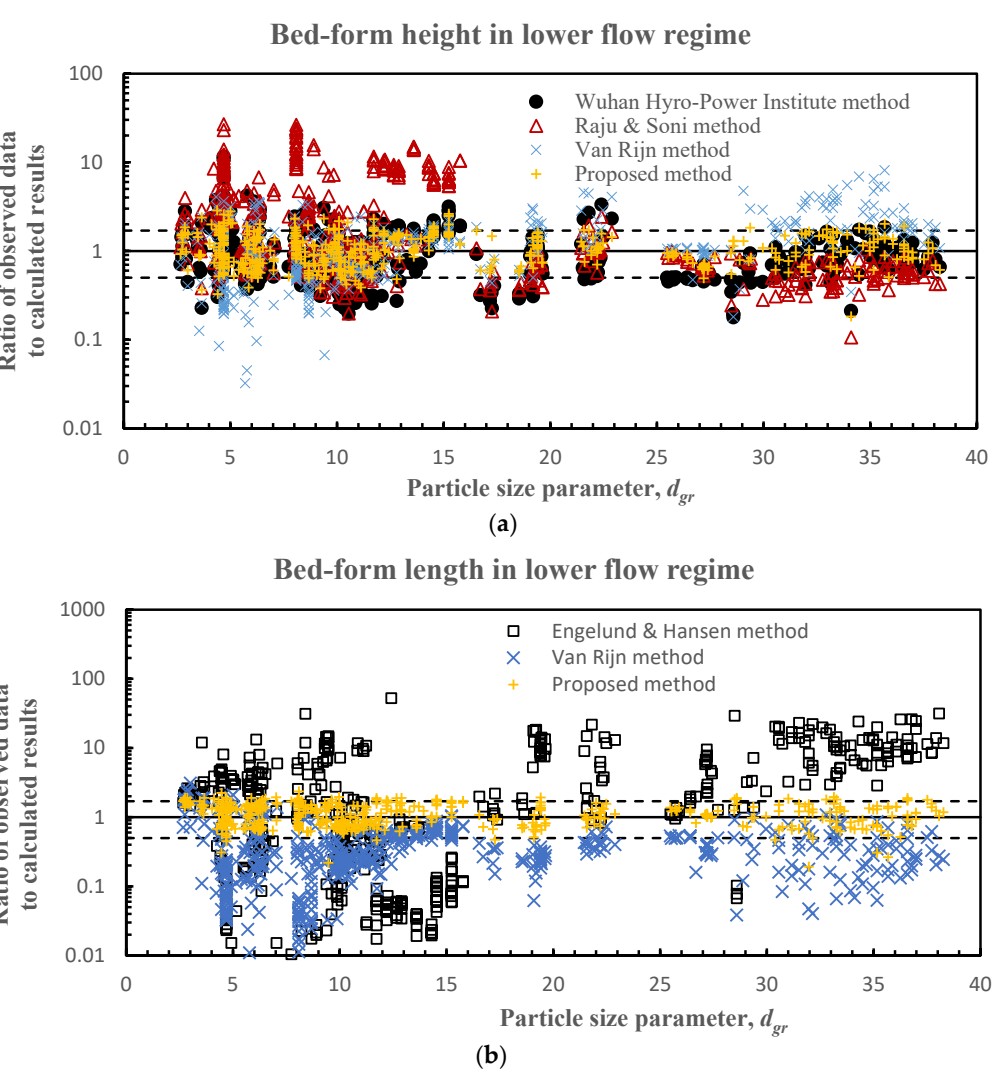

**Figure 13.** Comparison of bedform length in lower flow regimes using different methods. (**a**) Comparison of bedform height observed using different methods. (**b**) Comparison of bedform lengths observed using different methods.

Figures 12 and 13 show that there was always a large error when the dimensionless particle size was smaller than 20. The variations in bedform dimensions with *H/d* were analyzed based on the experimental and collected data. The calculation results of the typical models and the method proposed in this study were compared with the field-measured bedform height and length with different *H/d*. The trend curves of the calculated results and the field-measured data with *H/d* as the horizontal coordinate and the bedform dimensions as the vertical coordinate are shown in Figures 14–17. It is observed in these figures that the bedform height and length in different flow regimes increase significantly with an increase in *H/d*. The rate of increase in bedform dimensions with an increase in *H/d* increases when *H/d* is greater than $10^4$ in lower and upper flow regimes.

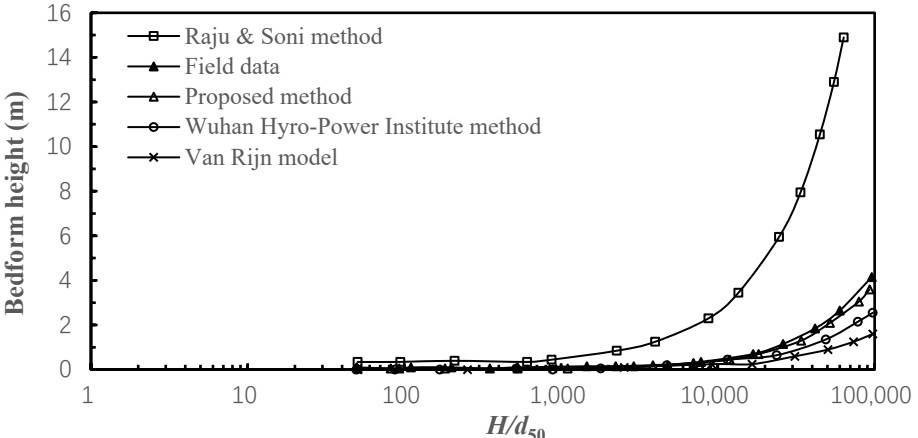

**Figure 14.** Relationship between $H/d_{50}$ and bedform height in lower flow regimes.

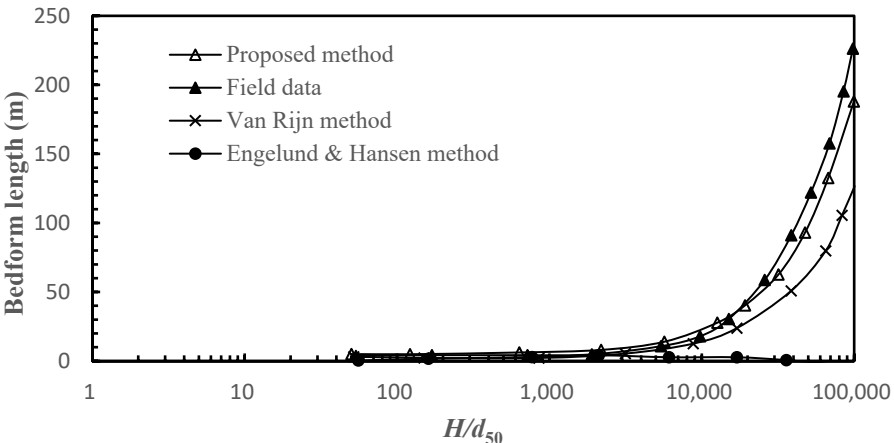

**Figure 15.** Relationship between $H/d_{50}$ and bedform length in lower flow regimes.

Figure 14 shows the trend curves of the calculated and field-measured bedform heights in lower flow regimes. The results calculated using the proposed method are the closest to the field-measured data. The results obtained using the Raju and Soni method have the greatest difference when *H/d* is greater than $10^3$. The variation trend of the calculated results is similar to that of the measured data.

Figure 15 shows the trend curves of the calculated and field-measured bedform lengths in lower flow regimes. The calculated results using the proposed method are the closest to the field-measured data. The results obtained using the Engelund and Hansen method have the greatest difference and cannot reflect the variation trend of the bedform length increasing with an increase in *H/d* when *H/d* is greater than $10^4$.

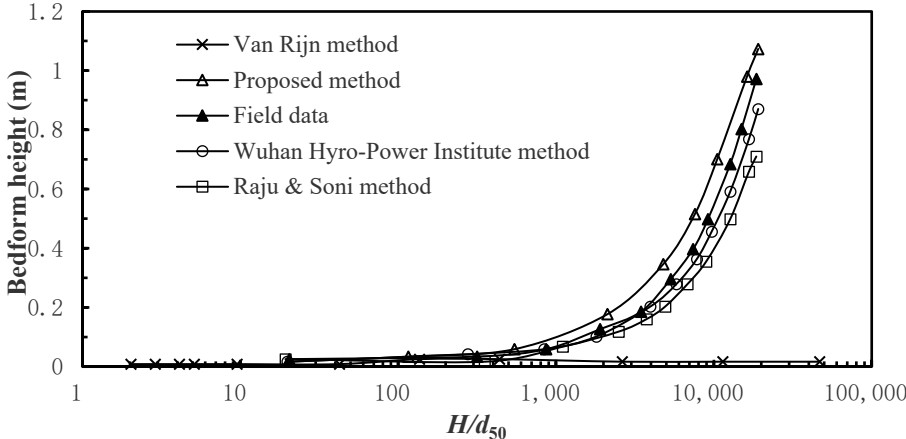

**Figure 16.** Relationship between $H/d_{50}$ and bedform height in upper flow regimes.

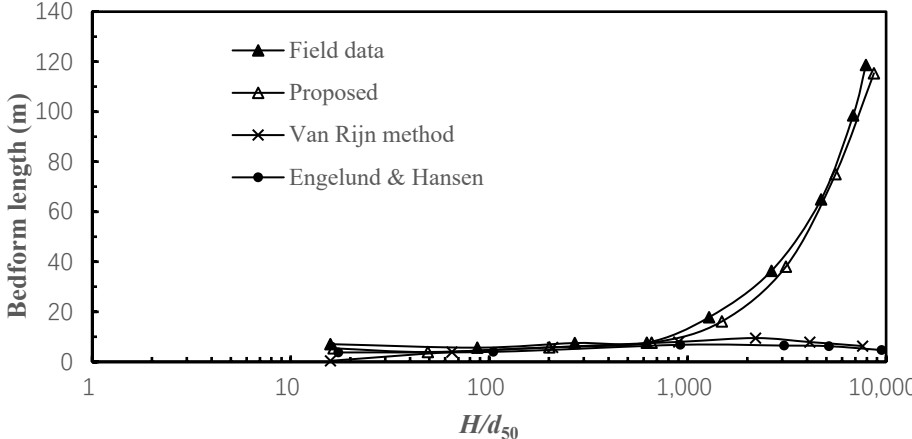

**Figure 17.** Relationship between $H/d_{50}$ and bedform length in upper flow regimes.

Figure 16 shows the trend curves of the calculated and field-measured bedform heights in an upper flow regime. The calculated results of the proposed method are the closest to the field-measured data. The results obtained using the van Rijn method are significantly different from the field-measured data and cannot reflect the variation trend of the bedform length increasing with an increase in $H/d$ when $H/d$ is greater than $10^3$.

Figure 17 shows the trend curves of the calculated and field-measured bedform lengths in the upper flow regimes. The calculated results of the proposed method are closest to the field-measured data. The results obtained by the van Rijn and the Engelund and Hansen methods are much different than the field-measured data and cannot reflect the variation trend of the bedform length increasing with an increase in $H/d$ when $H/d$ is greater than $10^3$.

Figures 14–17 verify that the momentum boundary-layer thickness may be a more suitable parameter than the flow depth to quantify the bedform dimensions on an alluvial bed. The Engelund and Hansen method produced the poorest prediction of bedform length. The Wuhan Hydropower Institute method produced a better prediction of bedform height than the van Rijn method and the Raju and Soni method.

## 4. Conclusions

In this study, the bedform dimensions on an alluvial bed with unidirectional flow were experimentally investigated, and the formulae for predicting the bedform dimensions on alluvial beds with unidirectional flow were derived by introducing a resistance coefficient for the flow in different regimes, momentum boundary-layer thickness, flow intensity, and hydraulic radius. A series of flume experiments was conducted, and 700 sets of

flume and field data were used to determine the coefficients of the formulae. Then, four typical formulae were used to compare the accuracy of the proposed formulae. The main conclusions are presented as follows.

The formulae for the prediction of the bedform dimensions on alluvial beds with unidirectional flow were proposed, and the momentum boundary-layer thickness was used instead of the flow depth as the main parameter. It was verified that the momentum boundary-layer thickness possesses a good correlation with bedform dimensions based on the experimental results and the collected data. The proposed formulae were more accurate than the other four typical formulae, especially when $H/d$ was greater than $10^3$.

The dimensionless sediment particle size should not be omitted in the calculation of bedform dimensions on alluvial beds with unidirectional flow, although the correlation between the dimensionless sediment particle size and bedform dimensions is weak, as reported by van Rijn (1984) [32]. The influence of dimensionless sediment particle size in the calculation of bedform dimensions in upper flow regimes when $\theta/\theta_{cr} \leq 7.5$ is particularly significant. In other cases, the products of the coefficient and the dimensionless particle size in the proposed formulae were within 2.

The bedform dimensions show an obvious trend of rapid increase with an increase in the ratio of flow depth to sand size ($H/d$), which is perfectly reflected by the proposed method. The bedform dimensions obtained using the van Rijn method and the Engelund and Hansen method were significantly different from the field-measured data and did not represent the variation trend of the bedform length in the upper flow regime. The Engelund and Hansen method yielded the poorest prediction for bedform length. The Wuhan Hydropower Institute method provided better prediction of bedform height than the van Rijn method and the Raju and Soni method.

**Author Contributions:** R.W.: Conceptualization, Methodology, Data curation, Formal analysis, Visualization, Investigation, Writing—original draft, Writing—review & editing. G.Y.: Supervision, Writing—review & editing, Project administration, Funding acquisition. All authors have read and agreed to the published version of the manuscript.

**Funding:** This research received no external funding.

**Data Availability Statement:** The data that support the findings of this study are available from the corresponding author upon reasonable.

**Conflicts of Interest:** The authors declare that they have no known competing financial interests or personal relationships that could have appeared to influence the work reported in this paper.

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
