# Peer review of "Prediction of Bedform Dimensions on Alluvial Bed in Unidirectional Flow"

_water, doi:10.3390/w16060893_

Round 1

Reviewer 1 Report

Comments and Suggestions for Authors

The authors propose a new formula for predicting the bedform height and length on an alluvial bed in a unidirectional flow, using laboratory experiments and field data. The analysis is an important one, bringing new contributions to the scientific community.

The performance of the proposed formula is analyzed compared to other known and usual formulas characteristic of such analyses, such as the van Rijn method, the Engelund and Hansen method, the Raju and Soni method, respectively the Wuhan Hydropower Institute method.

All parameters and coefficients of the proposed formula are calibrated for different analysis hypotheses: different flow regimes, different particle sizes, etc.

According to the results obtained, the calculations using the proposed formulas are more accurate and reasonable than those obtained with the usual formulas. The only possible disadvantage of the analysis is the fact that the analysis is not bi- or three-dimensional.

My decision is to accept the manuscript after the authors make some minor improvements regarding the quality and clarity of Figures 1a), 20, 21, 22 and 23. Minor corrections regarding the English language are also needed.

Comments on the Quality of English Language

Minor editing of English language required.

Reviewer 2 Report

Comments and Suggestions for Authors

The work concerns the prediction of the morphology of bedforms in an alluvial bed with a unidirectional flow from the experimental point of view. The authors showed for example that the dimension-less particle size should not be ignored in calculation of bedform dimensions, which is not a new finding. Moreover, the authors' experiment proofed the rapid increase with an increase in the ratio of flow depth to sand size (H/d), which was shown in many experiment in the past. Based only on the experimental data, the paper does not show any new data. However, it should be stated that the new formulae for predicting the bedform dimensions on alluvial beds in unidirectional flow were derived by introducing resistance coefficient for the flow in different regimes, momentum boundary-layer thickness, flow intensity and hydraulic radius and achievement of this goal is enough to make this paper publishable.

I have some general comments which in my opinion might make this paper more state-of-the art. It can be seen that the authors, have omitted several works in their references and, in my opinion, it would be good to add a little, at least a few items to the discussion or the introduction itself. As the paper is concerned with the morphology of benthic forms, it is surprising that no reference is made to the work of Coleman and Nikora, (2011) and whether a discrete or continuous approach was used. Here it is also worth noting other papers e.g. Bialik et al. (2014), which is associated with measurements of bed-forms dimensions in natural channel or Aberle et al. (2011) related to statistical investigation of bed-forms.

Also missing, is a clear reference to the classical approach presented by Ashley (1990) or Flemming (1988) and it would be useful for the paper to take such a reference into account, if, of course, the authors consider them appropriate.

On the other hand, the work is quite long and could be shortened and written more effectively. It seems that 27 figures are a lot and at least some of them could have been combined as sub-figure e.g. as (a), (b), (c)...

The paper also has editorial errors. According to the requirements of the mdpi, the literature should be in numerical order of appearance in the article. This rule was not followed, moreover some of the items do not have their number in the literature list, which shows that the paper was done on the fly.

References:

Bialik, R.J., Karpinski, M., Rajwa, A., Luks, B., Rowinski, P.M. (2014).  Bedform Characteristics in Natural and Regulated Channels: A Comparative Field Study on the Wilga River, Poland, Acta  Geoph. 62, 6, 1413-1434.

Coleman, S.E., and V.I. Nikora (2011), Fluvial dunes: initiation, characterization,
flow structure, Earth Surf. Process. Land. 36, 1, 39-57.

Comments on the Quality of English Language

The paper also has editorial errors. According to the requirements of the mdpi, the literature should be in numerical order of appearance in the article. This rule was not followed, moreover some of the items do not have their number in the literature list, which shows that the paper was done on the fly.

Reviewer 3 Report

Comments and Suggestions for Authors

The paper proposes a new equation to predict the bedform dimension based on the several factor like: the bed sediment particle size; the hydraulic radius associated with bed; the surface resistance; the effective flow intensity; the critical Shields number; the resistance coefficient for the flow in different regimes and the momentum boundary-layer thickness. The relation between bedform height and length with the parameters was analysed through experimental tests in two different water channels. From the acquired data, the equations coefficients were derived showing a R2 close to 1 with lowest value for the lower flow regime. At the end the proposed equation was compared with other experimental tests and other methodology proposed in literature showing a better prediction respect to other studies.

I think that the topic is interesting and in compliance with the scope of “Water Journal”. The paper achieved attractive results and the way of presenting the outcomes are generally good thus I suggest considering the manuscript through Minor Revision once the authors address the comments in the following:

·      The “Methods and materials” section should be improved. Two channels were used but only one is described and the second was only presented in the caption of figure 1 and the reader also need a description of tested flow condition. In addition the “Data collection” is not a part of obtained results but should be in the “Methods and material” section.

·      Please modify the year of the paper of Borkotoky, A. “Study on bedforms and lithofacies structures and interpretation of depositional environment of Brahmaputra River near Nemati, Assam, India”. In the References section is reported 2025 instead of 2015.

·      In the introduction section the authors should introduce the bedforms variation due to the sediment transport caused from the flow-vegetation interaction at different scale. The authors can refer to the study of Maio et Al 2022 “Impact of reconfiguration on the flow downstream of a flexible foliated plant” for the single plant scale; to the study of Zong & Nepf 2009 “Flow and deposition in and around a finite patch of vegetation” for the patch scale and to the study of Clary et Al “Sediment deposition and entrapment in vegetated streambeds” for the fully vegetated channel.

·      I suggest to merge the figures from 14 to 19 with the figures from 20 to 23 or at least the authors should show the results obtained with proposed methodology also in the figures with other studies (as they did from figure 24 to 27).

Round 2

Reviewer 2 Report

Comments and Suggestions for Authors

The work has been improved according to my comments. I think it is a better job now, although it should be emphasized that the novelty is not high. However, I think it will find an audience and has the potential to be quoted. After the corrections that have been made, I think it can be published in Water.